# Key Factors Influencing *Bacillus cereus* Contamination in Hot Ready-to-Eat Meal Delivery

**DOI:** 10.3390/foods14152605

**Published:** 2025-07-24

**Authors:** Tomáš Komprda, Olga Cwiková, Vojtěch Kumbár, Gabriela Franke, Petr Kouřil, Ondřej Patloka, Josef Kameník, Marta Dušková, Alena Zouharová

**Affiliations:** 1Department of Food Technology, Faculty of AgriSciences, Mendel University in Brno, Zemědělská 1, 61300 Brno, Czech Republic; olga.cwikova@mendelu.cz (O.C.); gabriela.franke@mendelu.cz (G.F.); petr.kouril@mendelu.cz (P.K.);; 2Department of Technology and Automobile Transport, Faculty of AgriSciences, Mendel University in Brno, Zemědělská 1, 61300 Brno, Czech Republic; vojtech.kumbar@mendelu.cz; 3Department of Animal Origin Food and Gastronomic Sciences, Faculty of Veterinary Hygiene and Ecology, University of Veterinary Sciences Brno, Palackého tř. 1946/1, 61242 Brno, Czech Republic; kamenikj@vfu.cz (J.K.); duskovam@vfu.cz (M.D.); skockovaa@vfu.cz (A.Z.)

**Keywords:** food safety, pathogenic bacteria, cooked rice, sugarcane bagasse, polypropylene

## Abstract

With increasing popularity of food delivery services, the microbial safety of transported meals should be ensured. An effect of the type of a meal (cooked rice; mashed potatoes; mushroom sauce), inner primary packaging (sugarcane bagasse [SB] tray; polypropylene [PP] tray), secondary container (polyester/polyethylene foam/aluminum foil [PPA] bag; PP box) on the time interval of the internal hot ready-to-eat (RTE) meal temperature decrease to the value critical for *Bacillus cereus* growth (40 °C) was tested during a simulated delivery; in aliquot samples of the same meals, *B. cereus* growth was quantified presuming a natural contamination of the meals. Type of a meal had no effect on the tested time interval (*p* > 0.05). Packaging a meal in the PP tray as compared to the SB tray and inserting primary trays into the PP box instead of PPA bag delayed (*p* < 0.05) the internal meal temperature decrease by 50 and 15 min, respectively. Average *B. cereus* counts in the naturally contaminated meals after the four-hour culturing at 40 °C was 2.99 log CFU·g^−1^. It was concluded that a hot RTE meal delivered up to four hours under the tested conditions is not likely to facilitate *B. cereus* growth above unacceptable levels.

## 1. Introduction

In recent years, popularity of food delivery services has grown significantly. Online food delivery platforms (Uber Eats, USA; Foodora, European Union; Meituan, China) have adopted market strategies to maximally reduce customer waiting times by informing restaurants about the food order and sending couriers to pick up the food from the restaurants and deliver it to the customers [1]. The delivery time is mostly not precisely specified, but the food should reach the consumer as soon as possible [2].

This draws increasing attention not only to the quality of the meals themselves but also to the conditions under which they are transported. One of the key factors influencing customer satisfaction is the temperature of the food upon delivery. This is not only a matter of sensory quality but also of microbiological safety [3]. While delivery speed plays an important role, the thermal performance of the packaging material is equally critical. Different types of packaging—ranging from plastic and aluminum to paper-based and biodegradable alternatives—exhibit varying insulation properties that can significantly affect the retention of food temperature [4,5,6].

Maintaining hot food at temperatures above 60 °C is essential to prevent it from entering the so-called “danger zone” (4 °C to 60 °C), where microbial growth occurs most rapidly [3]. Therefore, the ability of packaging to preserve thermal conditions for as long as possible is of paramount importance. Selecting appropriate packaging thus involves a trade-off between thermal efficiency, environmental impact, and chemical safety. For instance, certain plastic containers may leach endocrine-disrupting compounds such as bisphenol A (BPA) or phthalates when exposed to heat [4], while more sustainable materials often lack sufficient insulation, leading to faster cooling and potential quality degradation [7].

The present study focuses on analyzing the impact of packaging materials on the thermal stability of the hot ready-to-eat (RTE) meals during a prolonged (up to 4 h) delivery and evaluates how packaging choices can influence a microbial safety during transport. Counts of *Bacillus cereus* in the meal were chosen as a safety marker.

*B. cereus sensu lato* is a Gramm-positive, facultative anaerobic, motile, spore-forming, rod-shaped bacteria [8]; it is an opportunistic human pathogen causing gastrointestinal illnesses due to the ability of producing emetic (cereulide) and/or diarrheal toxins [9]. Emetic and diarrheal strains are able to grow at the pH range of 4.5–9.5, temperature range of 4–48 °C, minimal water activity of 0.93 and NaCl concentration up to 7%. Under the unfavourable conditions, *B. cereus* forms the heat resistant endospores [8]. As far as the toxins are concerned, diarrheal enterotoxins are heat-labile (inactivated by heating at 55 °C for 5 min), but cereulide, pre-formed in a food, remains stable at 121 °C for 2 h and can persist during food processing, including roasting and frying [10].

*B. cereus* infective dose, meaning a minimal concentration inducing a cereulide production is ambiguous, but the most frequently reported value is >10^5^ CFU·g^−1^ of food [11]. Guidelines regarding *B. cereus* counts usually mention levels <10^3^ CFU·g^−1^ as satisfactory, 10^3^–10^5^ CFU·g^−1^ as acceptable and >10^5^ CFU·g^−1^ as unsafe [12]. For a comparison, Juneja et al. [13] reported *B. cereus* counts in cooked rice stored 24 h at 45 °C to reach values of 10^6^–10^8^ CFU·g^−1^.

Due to the ability to form spores, it is practically impossible to prevent *B. cereus* contamination of foods [14]. *B. cereus* is ubiquitous and widespread in soil and water and consequently in roots and tubers of starch-rich plants; after the harvest, the spores are entering a food-processing equipment. Foods can be contaminated also via dust or insects [15]. However, according to Messelhäuser et al. [16], although *B. cereus* is widespread in an environment, the nature niches and the ways of entry into the food production and processing are largely unknown.

*B. cereus* is considered second most common etiologic agent of foodborne outbreaks in France and the third in the European Union [17]. Cereal products, rice, seeds dairy products, poultry, vegetables, herbs, spices and seafood are usually listed as the risky products [15], including RTE foods [18], soups and sauces [9]. Rice is mentioned as the most frequently contaminated food [19]. Fifty-eight % to 81% of rice samples in farms in Korea and 53% of rice samples sold in retail stores in the USA were contaminated by endospores of *B. cereus*, respectively [17].

The objective of the present study was to test following hypotheses. Regarding packaging materials for the hot RTE meal delivery, the tested null hypothesis presumed that type of a meal, packaging material (primary wrapping, secondary container, their arrangement) and external temperature during the delivery has no effect on the time interval of the meal internal temperature decrease to the risky value of 40 °C. As far as *B. cereus* is concerned, the following null hypotheses were considered: counts of *B. cereus* in naturally contaminated tested hot RTE meals do not exceed risky levels of 10^4^ colony forming units CFU·g^−1^ after four hours at the temperature of 40 °C; type of the hot RTE meal has no effect on *B. cereus* counts under these conditions.

## 2. Materials and Methods

### 2.1. Preparation of the Dishes

Three meals considered risky from the viewpoint of a microbiological safety (especially regarding sporogenic bacteria) were tested: cooked rice (composition: long-grain rice, onion); mashed potatoes (fresh potatoes, milk); mushroom sauce (dried mushrooms, beef broth, milk, cream, flour). All three meals were prepared according to respective recipes by Runštuk et al. [20]. Cooking temperature at least 90 °C was applied during meals production in order to substantiate their categorization as the hot ready-to-eat (RTE) meals. Testo 104-IR thermometer (Testo, Prague, Czech Republic) was used for verifying of the temperature.

Altogether 119 portions of mashed potatoes, 67 portions of cooked rice and 63 portions of mushroom sauce were prepared. Effect of selected conditions on the maintenance of the internal hot RTE meal temperature during a simulated delivery was evaluated using these successively prepared portions of each meal. From the randomly selected six portions of each type of a meal, an aliquot part was taken for evaluation of the *B. cereus* growth. Growth of *B. cereus* in this aliquot of the same meal was quantified in the laboratory (so, not directly during the simulated meal delivery, but under the corresponding temperature conditions; from the technical reasons it would be difficult to quantify *B. cereus* counts in the delivered meals after the each run of the simulated delivery).

### 2.2. Design of a Simulated Hot RTE Meal Delivery

A 150 g portion of each meal was put into each of the two tested primary wrappings (Figure 1): a polypropylene (PP) tray (100% polypropylene; Bittner Packaging Sp.j., Ożarów Mazowiecki, Poland) and a sugarcane bagasse (SB) tray (produced from the waste after industrial processing of sugarcane; Ecoware Solutions Private Limited, New Delhi, India), respectively. The trays were immediately heat-sealed (without any change of the atmosphere) with an overlying heat-sealing foil (polyethylene/polypropylene, 185 mm, thickness 52 μm; Maso-profit Ltd., Prague, Czech Republic), using a heat-sealing apparatus T-190 (MetalPack, Maso-profit, Ltd., Prague, Czech Republic) at the heat-sealing temperature of 180 °C. The sealing temperature was set according to the film supplier’s recommendations. The sealing time was only one second guaranteeing no deformation of the tray or the covering film.

Immediately after the heat-sealing with the overlying foil, the trays with the meals were inserted into each of the two types of the secondary containers (Figure 1). In order to simulate different conditions during the meal delivery, two types of secondary containers with different thermo-insulating properties were used: a three-layered bag consisting of a polyester external layer and two inner layers, polyethylene foam and aluminum foil (PPA bag; overall thickness of 17 mm; Guangzhou A.C.T. Products Co., Ltd., Guangzhou, China); a polypropylene thermobox (PP box; extruded polypropylene; thickness of 30 mm; Polibox Srl, Arluno, Italy).

The heat-sealed bowls were inserted into the secondary containers in one layer (altogether 2 trays), three layers (6 trays) and six layers (12 trays), respectively. A sensor of a datalogger (Extech SDL200; Teledyne FLIR, France) was stuck into each layer of trays (Figure 1e) and the internal temperature on each layer was recorded during a time interval from inserting the meals into the secondary container (time 0 min) to 420 min. The time interval of temperature decrease to the critical value of 40 °C was directly measured. A closed secondary container was exposed to the temperature conditions of a surrounding environment and the external temperature was recorded using a customary external thermometer. The simulation of the meal delivery proceeded during all seasons of the year in the range of external temperatures from −2 °C to +33 °C.

### 2.3. Microbiological Analysis

Microbiological safety of the hot RTE meals was assessed using aliquots of the same meals, in which *B. cereus* growth was measured in laboratory conditions using a protocol by [21]. In the quoted experiment, the authors monitored growth of *B. cereus* after inoculation of the meals with spores of an emetic and diarrheic strain, respectively. In the present study, a natural contamination of the meals with *B. cereus sensu lato* (irrespective of a strain) was evaluated.

Six 150 g portions of each meal prepared as mentioned above were inserted into the PP trays (sugarcane bagasse trays were not used in this part of the experiment, only PP trays). The trays were immediately heat-sealed as mentioned above and then two trays of each meal were incubated (INCU-Line; VWR, Radnor, PA, USA) for 0, 0.5, 1.0, 1.5, 2.0, 2.5, 3.0, 3.5 and 4.0 h at 40 °C, 50 °C and 60 °C, respectively. Subsequently, 25 g of each sample was weighed into sterile homogenization bags and samples were diluted 1:9 with buffered peptone water and homogenized in a stomacher (Star Blender LB 400; VWR, Radnor, PA, USA) for 90 s. The homogenate was inoculated on Mannitol Yolk Polymyxin (MYP) agar (Oxoid, Thermo Fisher Scientific, Brno, Czech Republic) and cultivated for 24 h at 30 °C.

Suspected colonies of *B. cereus* were reinoculated onto blood agar (OXOID) and isolates showing a complete hemolysis were identified by Matrix-Assisted-Laser-Desorption-Ionization-Time-of-Flight mass spectroscopy (MALDI-TOF MS). Individual colonies of suspected *B. cereus* isolates were spread in duplicate on a MALDI plate, overlaid with 1 μL of HCCA matrix (a saturated solution of α-cyano-4-hydroxycinnamic acid in 50% acetonitrile, 47.5% water and 2.5% trifluoroacetic acid; Merck KGaA, Darmstadt, Germany) and then overlaid with 1 μL of 70% formic acid. The matrix with the colonies was let dry and after drying were isolates analyzed by MALDI-TOF MS (UltraFleXtreme instrument, Bruker Daltonik, Bremen, Germany; FlefControl 3.4 software; BioTyper 3.0 software, Bruker Daltonik; BioTyper database entries, version 10.0). Identification at the species level was considered reliable at the BioTyper log(score) > 2.0.

For species identification of the suspected *B. cereus* sensu lato isolates, that were not identified as *B. cereus* sensu stricto by a polymerase chain reaction (PCR; see below), the following collection strains were used for confirmation of the objectivity of the identification: *B. cereus* DSM 4312, *B. cereus* CCM 2010, *B. cereus* CCM 869, *B. thuringiensis* CCM 19, *B. mycoides* CCM 115, *B. weihenstephanensis* CCM 4872, *B. cytotoxicus* DSM 22905.

*B. cereus* sensu stricto isolates were confirmed by PCR. Samples of bacterial colonies grown on plate count agar (TGYE: tryptone, glucose, yeast extract) 24 h at 30 °C were suspended in 100 μL of sterile physiological solution, heated 10 min at 100 °C and centrifuged at 2082× *g* for 1 min. The supernatant was transferred to a new centrifuge tube and an aliquot of 2 μL was used as template DNA in the PCR. The *gyrB* gene coding for DNA gyrase subunit B (primers BC1 [5ʹ-ATTGGTGACACCGATCAAACA-3ʹ] and BC2 [5ʹ-TCATACGTATGGATGTTATTC-3ʹ]) was used for *B. cereus* sensu stricto identification. The amplified products were separated on a 2% agarose gel in 0.5 × TBE buffer. The gels were stained with Midori green (Nippon Genetics, Düren, Germany) and visualized by a UV transilluminator (VWR, Radnor, PA, USA).

### 2.4. Statistical Evaluation

Normality of the data distribution was evaluated by Kolmogorov-Smirnov test. The data regarding effects of selected external factors (type of a dish; material of a primary wrapping; type of a secondary container; number of layers of the primary bowls stored in the secondary container; external temperature) on the time interval of the internal dish temperature decrease during a simulated delivery were assessed by factorial ANOVA. The time interval of the dish internal temperature decrease to 40 °C (risky value for the *B. cereus* growth) was used as a dependent variable (irrespective of the fact that the dish temperature was recorded until the time of 240 min). As far as external temperature is concerned, for the purpose of factorial ANOVA, not a numerical sequence, but five temperature spans (very low: <0 °C; low: 0–10 °C; intermediate: 10–20 °C; high: 20–30 °C; very high: >30 °C) were used as a one of the independent variables.

Differences in the time intervals of the internal dish temperature decrease between the dishes, between the primary wrappings, between the secondary containers and between the numbers of layers, respectively, were evaluated by one-way ANOVA with *post-hoc* Tukey’s test.

Regression analysis was used for evaluating a dependence of the time interval of the internal dish temperature decrease on the external temperature during the simulated delivery, and for evaluating a dependence of *B. cereus* counts on the time interval of the internal dish temperature decrease; in this case, an uninterrupted sequence of all measured numerical temperature values was applied. 

Distribution of the microbiological data was not normal based on Kolmogorov-Smirnov test. So, the differences between counts of *B. cereus* strains (including control representing inadvertent contamination of a dish) growing at three tested temperatures (60 °C; 50 °C; 40 °C) and in the three different dishes (cooked rice; mashed potatoes; mushroom sauce), respectively, were evaluated by a nonparametric Kruskal-Wallis test with *post-hoc* Nemenyi test.

Statistica 14 software (TIBCO Software Inc., Santa Clara, CA, USA) was used for all statistical evaluations.

## 3. Results

Effects of the selected variability factors (type of a dish; material of a primary wrapping; type of a secondary container; number of layers of the primary bowls stored in the secondary container) on the time interval of the internal dish temperature decrease from the original temperature (an instant of inserting the tray with the tested meal to the secondary container) to 40 °C during a simulated delivery are shown in Table 1. Type of a dish had no significant effect on the tested time interval (Tukey’s test; *p* > 0.05). Material of a primary tray and of a secondary container altogether accounted for 20% (12 + 8) of explained variability (Tukey’s test; *p* < 0.001). External temperature during the meal delivery and number of the tray layers in a secondary container were the most important factors determining the time interval of the internal dish temperature decrease to the critical value of 40 °C: 47% and 21% of explained variability, respectively (Tukey’s test; *p* < 0.001).

The data shown in Figure 2 fully correspond to the results presented in Table 1. No differences in the tested time interval between dishes (Tukey’s test; *p* > 0.05) were found out. The data presented in Figure 2 for a given independent variable were calculated (summed up) irrespective of all other independent factors. So, in the case of a dish, the time of the temperature decrease to 40 °C ranged between 100 and 110 min (Tukey’s test; *p* > 0.05), calculated irrespective of primary wrapping, secondary container and number of layers.

On the other hand, significant differences (Tukey’s test; *p* < 0.05) between primary tray materials and between types of a secondary container, respectively, were established. Inserting a meal into a polypropylene tray instead of a sugarcane bagasse tray prolonged a temperature decrease to 40 °C by nearly 50 min (Figure 2) and putting the primary trays into polypropylene box prolonged the temperature decrease by approximately 15 min as compared to the polyester/polyethylene foam/aluminum foil bag.

Most importantly, the time of the internal dish temperature decrease to 40 °C (calculated irrespective of the type of a dish, type of a primary tray or material of a secondary container) was approximately 45 min when only one layer of trays was put into a secondary container, but 100 min when three layers were applied, and 140 min in the case of six layers (Tukey’s test; *p* < 0.05; Figure 2).

The time of the internal dish temperature decline to 40 °C decreased linearly with decreasing external temperature during the simulated hot dish delivery (Figure 3). The dependence was highly significant (*p* < 0.001; due to a high number of measurements, n = 250), but at the same time coefficient of determination of the relationship was very low (R^2^ = 0.12; high variability of the measured data).

In the microbiological part of the present experiment, the time of the internal dish temperature decline to 40 °C, which was used as a dependent variable in Figure 3, was employed as an independent variable for an evaluation of the dependence of *B. cereus* growth on the time of the hot ready-to-eat meal delivery (as already mentioned in the Materials and Methods section, *B. cereus* counts were not measured directly in the meals tested during the simulated delivery, but at the laboratory conditions in the aliquot parts of the same three meals).

The results are shown in Figure 4. The dependence was significantly better (*p* < 0.01) described by an exponential regression than by a linear one. The time axis in Figure 4 is expressed in hours, because the linear and quadratic term of the regression equation would be very low (y = 1.43 − 0.003x + 0.00004x^2^) when calculated in minutes (as in Figure 3). Moreover, the number of points shown in Figure 4 is low as compared to the total number of measurements due to the overlapping of many nearly identical values. As it is apparent from Figure 4, counts of *B. cereus* in naturally contaminated meals after the four-hour culturing at 40 °C reached an average value of 3 log CFU·g^−1^ of meal (evaluated according to the regression that was calculated irrespective of the tested meal). However, the *B. cereus* counts reached nearly 4 log CFU·g^−1^ in some cases, which also confirms the data shown in Figure 5.

*B. cereus* counts in all three tested meals were below the limit of detection (<1.7 log CFU·g^−1^) before incubation. As follows from Figure 5, the culturing temperature was able to affect *B. cereus* counts. However, the counts at 40 °C were significantly higher (Nemenyi test; *p* < 0.05) in comparison with both 60 °C and 50 °C only in mashed potatoes. Only tendency (Nemenyi test; *p* = 0.09) was found out in cooked rice, and *B. cereus* counts were practically the same in mushroom sauce cultured in the all three temperatures (Nemenyi test; *p* > 0.05). As far as the comparison of the meals is concerned, no differences in *B. cereus* counts between mashed potatoes, cooked rice and mushroom sauce cultured at 60 °C were established (Nemenyi test; *p* > 0.05). The same was true regarding the temperature of 50 °C (Nemenyi test; *p* > 0.05). On the other hand, as far as the temperature of 40 °C is concerned, *B. cereus* counts in mashed potatoes was higher (Nemenyi test; *p* < 0.05) as compared to mushroom sauce and tended to be higher (*p* = 0.06) in comparison with cooked rice. Similarly, only tendency (Nemenyi test; *p* > 0.05) to higher *B. cereus* counts in cooked rice in comparison with mushroom sauce was established (Figure 5).

## 4. Discussion

The negligible effect of a meal on the time interval of the internal meal temperature decrease (*p* > 0.05; Figure 2) is explained by the high-water content in all three dishes, and water has one of the highest specific heat capacities (approximately 4186 J∙kg^−1^∙K^−1^). The mushroom sauce tended to reach higher time interval, as it contains more water, approximately 90 g per 100 g of dish, which is mainly due to the content of the main ingredients (beef broth, milk, cream). As reported by [22,23], the specific heat capacity of mashed potatoes and cooked rice is similar, reaching values of 3400 and 2700 J∙kg^−1^∙K^−1^, respectively.

Two materials were used for primary food packaging in our study: conventional PP and an advanced material made from sugarcane waste, known also as sugarcane bagasse (Figure 1c,d). Bagasse properties can be found out in several studies [24,25], including its thermal properties [5] and its use in packaging materials [26]. According to these studies, bagasse consists of approximately 40–50% cellulose, 25–35% hemicellulose, 15–20% lignin and a low percentage of ash. According to [27], the thermal conductivity of bagasse is around 0.05 W∙m^−1^∙K^−1^ (dry) and 0.11 W∙m^−1^∙K^−1^ (wet), which values are likely to increase at temperatures of 40–90 °C (the study of [27] deals with sweet sorghum bagasse, but it is assumed that the thermal properties of sugarcane bagasse used in the present study are very similar).

Nevertheless, the thermal conductivity of bagasse is still lower in comparison with PP (0.2 W∙m^−1^∙K^−1^), which implies better thermal insulation properties of bagasse. This is contrary to the results of our study, where the internal temperature of the meal wrapped in the sugarcane bagasse tray and the PP tray decreased to 40 °C within an average time interval of 60 min and 110 min, respectively (Figure 2). Although these results could be partially biased by the unequal numbers of both types of the primary packaging used in the present study (the number of the respective meals put into the sugarcane bagasse trays was nearly ten times lower in comparison with the PP trays: 24 and 225 trays of sugarcane bagasse and PP were used, respectively), our findings do not confirm superior thermal properties of bagasse [27].

This result, on the other hand, does not question environmental benefits of bagasse. In terms of sustainability, sugarcane bagasse trays are clearly superior. Bagasse (formerly burned as a waste) is nowadays used to make eco-friendly packaging and disposable tableware, which are the typical examples of upcycling, i.e., using waste to make a new product [28]. Another advantage of bagasse-based products is their compostability [29]. According to the EN 13432:2000 standard [30], bagasse decomposes in 45–60 days in an industrial composting plant and in a matter of months in a home compost. On the other hand, PP packaging decomposes hundreds of years [31].

When considering a choice of a proper packaging material for the hot RTE meal delivery, their toxicological attributes should also be taken into account (though we did not consider them in the present study). Regarding the food safety concerns, sugarcane bagasse trays do not contain toxic substances and are safe from the viewpoint of contact with food, whereas residual toxic contaminants, such as aldehydes, ketones, aromatic hydrocarbons, plasticizers, etc., were reported to be released from PP packaging into food [32]. PP may also release chemical substances such as bisphenol A at elevated temperatures [7]. Regarding thermal conductivity of PP, Zemanová [4] reported its ability to maintain its temperature above 40 °C for 30–45 min.

As far as secondary containers are concerned, the two types of packaging usually employed by food delivery companies were used in the present study (Figure 1a,b), and significant difference between their abilities to maintain internal temperature of the transported meal above 40 °C was demonstrated (*p* < 0.05; Figure 2). Polypropylene thermobox prolonged the temperature decrease by 15 min compared to secondary container consisting of polyester/polyethylene foam/aluminium foil. However, it should be noted that the PP box has a wall thickness of 35 mm, while the PPA bag, though three-layered, was only 17 mm thick. With comparable material thicknesses, the layered packaging would be more suitable, as the thermal conductivity of its sandwich wall is 4–6 times lower (0.035–0.05 W∙m^−1^∙K^−1^) than that of pure PP (0.2 W∙m^−1^∙K^−1^; [33,34]).

As far as the filling of the space of a secondary container is concerned (see Figure 1e), the significant differences in the time interval of the internal meal temperature decrease to 40 °C between numbers of layers of the primary bowls stacked in secondary container (*p* < 0.05; Figure 2) should be expected: total heat (Q) is directly proportional to the total mass of the substance m (in this case mass of the meal), specific heat capacity (c) and the temperature difference (∆T) according to the equation Q = m ∙ c ∙ ∆T [35]. The more food is there in the secondary container, the higher is the specific heat capacity of the entire internal system, because the specific heat capacity of the meal is always higher than that of air (1000 J∙kg^−1^∙K^−1^).

As far as the health risk presented by the growth of *B. cereus* is concerned, three independent variables were considered in the present study: culturing temperature (40 °C), maximum time of the simulated hot RTE meal delivery (four hours) and a type of the RTE meal, respectively. Consequently, two hypotheses were tested: *B. cereus* counts in naturally contaminated RTE meals do not exceed unacceptable levels after the 4-h culturing (and relatedly during the 4-h simulated delivery) at the temperature of 40 °C; type of the RTE meal has no effect on *B. cereus* counts under these conditions (4-h/40 °C culturing).

Three naturally contaminated model RTE meals were used in the present study, which is different in comparison with other similar experiments, where the tested foods were usually inoculated with the heat-shocked *B. cereus* spores [13,21,36,37].

A threshold, below which the inner temperature of the hot RTE meals should not decrease, was fixed to 40 °C in the present study (and the pertinent time interval to this point was measured; Figure 4). In this context, a value of 30–40 °C is often mentioned as an optimum for *B. cereus* growth [13,36], maximum growth temperature being 55 °C [13]. On the other hand, according to [38], a sufficient temperature, where *B. cereus* spores are not able to germinate and vegetative forms cannot multiply, is ≥60 °C. Similarly, based on the current legislation [2], finished food should be kept at a temperature of at least 60 °C after cooking.

Hot RTE meals are distributed in such a way as to reach the consumer as quickly as possible at a temperature of at least 60 °C (requirement of the Czech law [2]), which also corresponds to the Food and Drug Administration requirements for maintaining hot foods at temperatures of 140 °F or higher [39]. The delivery of hot RTE meals by courier services has become widespread in large cities in Europe in recent years. While a minimum temperature of 60 °C can be achieved when hot dishes are distributed in catering service establishments, this is practically impossible when hot RTE meals are transported over longer distances, which is associated with longer time intervals and, consequently, with a risk of the growth of undesirable bacteria, including *B. cereus*. Therefore, irrespective of the fact that the upper limit of the growth temperature for *B. cereus* was reported to be 48 °C [13], the temperatures of 40 °C, 50 °C and 60 °C were chosen in the present study as the model temperatures from the following reasons: 60 °C as the limit temperature given by regulations, and temperatures of 40 °C or 50 °C as the values at which the growth of *B. cereus* sensu lato is still possible.

The above-mentioned value of 60 °C is usually based on the often-reported threshold value for *B. cereus* counts of 5 log CFU·g^−1^ of food [15]. However, the most important threat for human health as far as *B. cereus* is concerned are not its counts as such, but a production of the emetic toxin cereulide [9]. We did not measure *B. cereus* toxins in the present study, but, when discussing a critical temperature, it should be mentioned that cereulide production seems to be restricted to the temperature range of 12 °C to 40 °C [40,41]). Based on the current scientific knowledge, cereulide production stops at 40 °C [9]. Krenzler et al. [42]) detected no cereulide at this temperature. Finlay et al. [41] reported a drastic decrease of cereulide production when temperature exceeded 40 °C despite the fact that *B. cereus* still grew at a fast rate. It follows from the study of [43] that the cell numbers do not reliably correspond with cereulide toxin production and are therefore not a suitable indicator.

So, it could be concluded that the temperature of 40 °C can be considered a satisfactory lower limit for handling the hot RTE meals as far as *B. cereus* is concerned.

Another factor tested in the present study was the time interval of the decrease of the original internal hot RTE meal temperature to the above-discussed value of 40 °C during the simulated meal delivery. Current legislative concerning meal delivery does not explicitly specify the time, only stipulates that the meals should reach the consumer as quickly as possible [2]. In the present study, the maximum time of 4 h was chosen as an independent variable concerning the hygienic risk of the presence of *B. cereus* in a meal. Although a time interval of the RTE meal delivery is usually much shorter, for the sake of safety, the time of 4 h was considered as a worst-case scenario.

It is worthy to mention that there could be an opposite problem during the meal delivery at the low ambient temperatures, when the internal dish temperature could decrease to the critical value of 40 °C within the time interval as short as one hour (Figure 3). However, *B. cereus* counts in the tested model dishes in the present study did not exceed 1.5 log CFU·g^−1^ as far as this time interval is concerned, but reached 3 log CFU·g^−1^ after 4 h (based on the pertinent regression equation; Figure 4). The experiment was carried out during all seasons of a year within a broad external temperature range (from –2 °C to +33 °C). So, it can be concluded in this context that the results of the present study are applicable for both freezing and hot conditions within different seasons and different climate zones.

It is undeniable that variable external temperatures during delivery can affect microbial safety of the hot RTE meals. However, most hot meals are transported over relatively short distances in thermoboxes using, among other things, bicycles, e-scooters and similar contemporary vehicles. In this case, the temperature conditions are almost identical to the simulated delivery used in the present study, which was demonstrated as safe. If e.g., passenger cars are used for transportation in the colder months, the temperature conditions inside the thermoboxes are again similar to simulations in the laboratory surroundings applied in the present experiment for *B. cereus* culturing.

Nevertheless, a decrease of the internal temperature of the hot meals can still occur, which is also apparent from the results of the present experiment. When taking into account the key drop of temperature below 50 °C and the time interval of the meal delivery under the above-mentioned conditions, our study demonstrated safety of the transport itself: the time interval is not sufficient in the most cases to exceed the lag phase of the *B. cereus* growth. More important from the viewpoint of the food safety is the behaviour of a consumer after the hot RTE meal has been delivered: how quickly is the meal consumed or under which conditions it is stored until consumption.

As far as *B. cereus* counts in the RTE foods are concerned, 10^3^–10^5^ CFU·g^−1^ or ml are considered acceptable, the value exceeding 10^5^ already unsafe [12]. Other studies [44] report values > 10^4^ CFU·g^−1^ or ml as unsafe regarding bulk RTE foods containing rice (one of the meals used in the present study). Soares et al. [45], when classified microbiological quality of university canteens (in northern Portugal), reported counts of *B. cereus* < 2 log CFU·g^−1^ as satisfactory, ≥2 ≤ 3 log CFU·g^−1^ as acceptable, >3 < 5 log CFU·g^−1^ as unsatisfactory and ≥5 CFU·g^−1^ as unacceptable.

Comparison of our results regarding *B. cereus* counts after the 4-h culturing (relatedly during the 4-h simulated delivery; Figure 2, Figure 3 and Figure 4) with the literature data is difficult, because only a few studies explicitly mention delivery of RTE cooked foods [46]. Mass catering [47], restaurants, cafés, catering and takeaway food premises [48] or university canteens [45] are evaluated instead. In comparison with the schedule used in the present study (the time limit for the meal delivery of four hours, the lower temperature limit of 40 °C), the authors apply lower target temperatures (24 °C, [37]; 18 °C, [49]), but often longer culturing intervals (up to 24 h, [50]) or use heat-shocked *B. cereus* spores [13] instead of natural contamination of meals (the present study).

Based on the type of a meal, inner primary wrapping, outer secondary container and number of layers of the primary trays in the secondary container tested in the present study, the time of the internal RTE meal temperature decrease to 40 °C was in the range of 60 to 140 min (Figure 2), with corresponding *B. cereus* counts of 1.40 log CFU·g^−1^ to 1.80 log CFU·g^−1^ (Figure 4).

When taking into account the above-mentioned caveats, at least rough comparison of our results with the literature data can be inferred. In an experiment of [36], *B. cereus* counts in rice (one of the dishes tested also in the present study) inoculated by the spore cocktail increased from 2.15 log CFU·g^−1^ to 2.41 log CFU·g^−1^ after 6 h of exponential cooling from the initial temperature of 54.5 °C; a cooling time of approximately 3 h from 54.5 °C to ca. 40 °C can be extrapolated from the data of [36]. Juneja et al. [13]), reported *B. cereus* counts in cooked rice to be 4 log CFU·g^−1^ after 4 h of storing at 40 °C. This value is by 2 log CFU·g^−1^ higher in comparison with our data as far as cooked rice is concerned (Figure 5), probably due to the fact that the quoted authors inoculated the tested dish with the heat-shocked spores (natural contamination was evaluated in the present study).

According to [37], the lag phase of the *B. cereus* growth in cooked rice stored at 24 °C was 2–3 h and the storage time required to multiply *B. cereus* to the critical levels of 3 log CFU·g^−1^ was 5 h and 4.7 h regarding white and brown rice, respectively. Ankolekar and Labbé [51]) reported *B. cereus* counts in rice stored for 4 h at 20 °C to be 2.24 log CFU·g^−1^. The data are only seemingly similar to our results regarding rice (Figure 5), because both temperature differences (20 °C vs. 40 °C in our study) and the fact that the quoted authors tested the spore-inoculated samples should be taken into account.

Apart from cooked rice, the literature data regarding other two meals tested in the present study (mashed potatoes, mushroom sauce) are rather scanty as far as *B. cereus* growth is concerned. From the viewpoint of comparable time/temperature values, no useful data are available regarding hot RTE mushroom sauce delivery. As far as thermally treated potato puree is concerned, it is possible to extrapolate from the results of [52] an increase of *B. cereus* counts from 3.5 log CFU·g^−1^ to 4.0 CFU·g^−1^ in the spore-inoculated samples stored at 20 °C for 4 h. This is similar to the *B. cereus* counts in mashed potatoes cultured at 40 °C in the present study (Figure 5).

## 5. Conclusions

The null hypothesis presuming a negligible effect of a type of the tested hot RTE meal on the time interval of the dish internal temperature decrease to the value risky from the viewpoint of *B. cereus* growth (40 °C) was confirmed: no differences between mashed potatoes, cooked rice and mushroom sauce was found out from this viewpoint. On the other hand, both inner primary packaging and outer secondary container significantly affected the time of keeping the temperature above the threshold value. Contrary to the expectation, the presumed better thermal properties of the sugarcane bagasse as compared to polypropylene (PP) was not confirmed in the present study (Figure 2).

The lower time interval for the maintenance of the inner meal temperature when the dishes were put into the secondary container composed of polyester/polyethylene foam/aluminum foil (PPA) in comparison with the PP box was likely only a consequence of the lower wall thickness of the PPA bag, because the thermal conductivity of this sandwiched wall is several times lower than that of the PP box.

As far as *B. cereus* is concerned, the null hypothesis that counts of this pathogen in naturally contaminated hot RTE meals do not exceed unacceptable levels (10^4^ CFU·g^−1^) after the 4-h culturing (and relatedly during the 4-h simulated delivery) at the temperature of 40 °C was confirmed: based on the regression equation presented in Figure 4, the 4-h time interval corresponds to *B. cereus* counts of 2.99 log CFU·g^−1^. On the other hand, the hypothesis that the type of the hot RTE meal has no effect on *B. cereus* counts under these conditions (4-h/40 °C culturing) was not corroborated: *B. cereus* counts were higher in mashed potatoes in comparison with mushroom sauce.

It could be concluded that a hot RTE meal delivered up to four hours under the conditions tested in the present study is not likely to facilitate *B. cereus* growth above the unacceptable levels and so endanger a consumer. The safety conclusions are somewhat limited by the absence of quantification of *B. cereus* toxins (especially emetic toxin cereulide). Detection of the toxins and of the relevant virulence genes in isolated *B. cereus* strains is one of the objectives of the authors of the present study regarding further experiments in this field.

## Figures and Tables

**Figure 1 foods-14-02605-f001:**
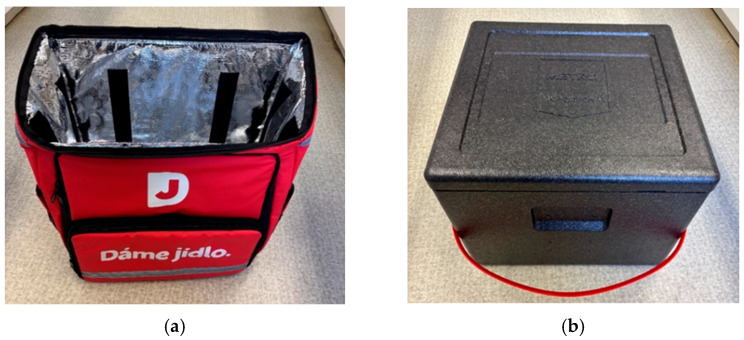
Types of secondary containers and primary wrappings used in a simulated dish delivery; (**a**): polyester/polyethylene foam/aluminum foil bag (thickness of 17 mm; Guangzhou A.C.T. Products Co., Ltd., Guangzhou, China); (**b**): polypropylene thermobox (extruded polypropylene, thickness of 30 mm; Polibox Srl, Arluno, Italy); (**c**): polypropylene tray (Bittner Packaging Sp.j., Ożarów Mazowiecki, Poland); (**d**): sugarcane bagasse tray (Ecoware Solutions Private Limited, New Delhi, India); (**e**): internal temperature measurement using a puncturing sensor of a datalogger (Extech SDL200, Teledyne FLIR, France).

**Figure 2 foods-14-02605-f002:**
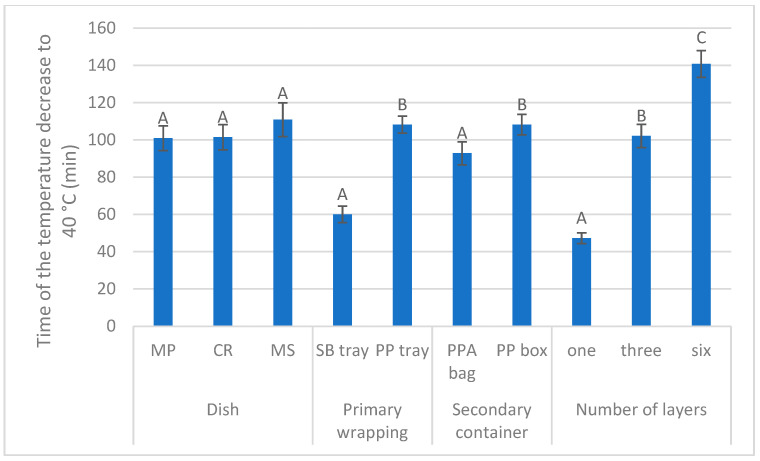
Effect of a dish (mashed potatoes, MP, n= 119; cooked rice, CR, n = 67; mushroom sauce, MS, n = 63); inner primary wrapping (sugarcane bagasse [SB] tray, n = 24; polypropylene [PP] tray, n = 225); outer secondary container (polyester/polyethylene foam/aluminum foil [PPA] bag, n = 75; polypropylene [PP] box, n = 174); and number of layers of the primary trays in the secondary container (one layer, n = 61; three layers, n = 92; six layers, n = 96) on the time interval of the internal dish temperature decrease to 40 °C; the data regarding a given independent variable were calculated irrespective of other independent variables; A–C: means with different superscripts within a given independent variable differ at *p* < 0.05 (one-way ANOVA with *post-hoc* Tukey’s test).

**Figure 3 foods-14-02605-f003:**
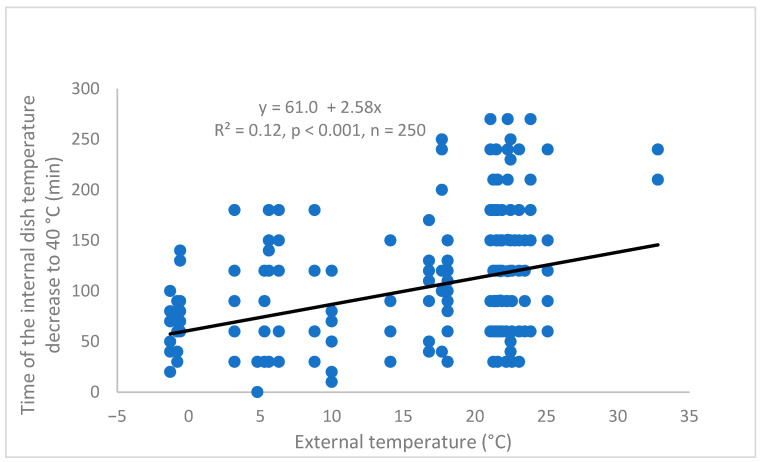
Dependence of the time interval of the internal dish temperature decrease from the original temperature (an instant of inserting the tray with the tested meal to the secondary container) to 40 °C on the external temperature during a simulated ready-to-eat dish delivery; the regression was calculated irrespective of a dish, inner primary wrapping, secondary container and number of layers of the primary trays in the secondary container, respectively.

**Figure 4 foods-14-02605-f004:**
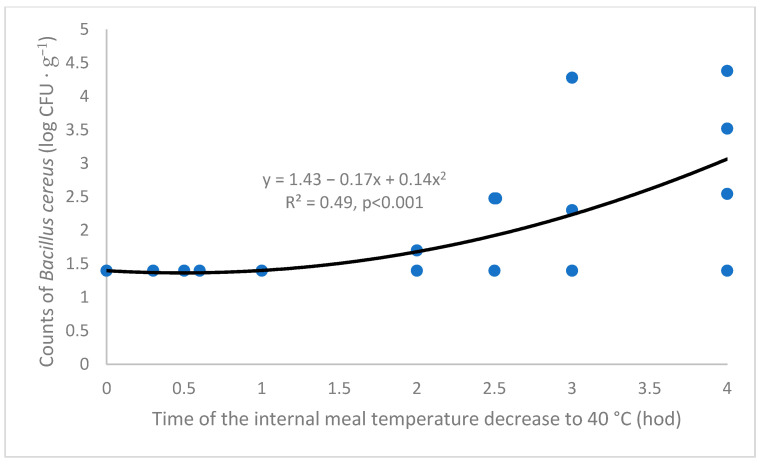
Dependence of *Bacillus cereus* counts in the three naturally contaminated model meals (cooked rice, mashed potatoes, mushroom sauce) on the time interval of the internal meal temperature decrease from the original temperature (an instant of inserting the tray with the tested meal to the secondary container) to 40 °C; the values of the independent variable were calculated irrespective of a meal, inner primary wrapping, secondary container and number of layers of the primary trays in the secondary container; the values of the dependent variable were calculated irrespective of the meal, n = 172.

**Figure 5 foods-14-02605-f005:**
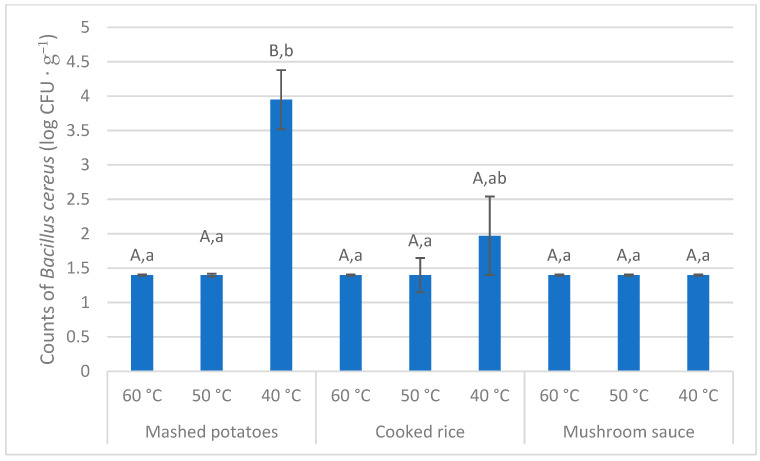
Counts of *Bacillus cereus* in the naturally contaminated tested meals after the 4-h culturing at three different temperatures; A,B—means with different superscripts within a given meal differ at *p* < 0.05; a,b—means with different superscripts within a given culturing temperature differ at *p* < 0.05; Kruskal-Wallis test with *post-hoc* Nemenyi test (n = 2).

**Table 1 foods-14-02605-t001:** Effect of the tested variability factors on the time interval of the internal dish temperature decrease from an instant when the tray with the tested meal was inserted to the secondary container to the critical value of 40 °C; factorial ANOVA, n = 180 (three dishes [mashed potatoes, cooked rice, mushroom sauce] × five temperature spans [very low: <0 °C; low: 0–10 °C; intermediate: 10–20 °C; high: 20–30 °C; very high: >30 °C] × two primary wrappings [polypropylene tray; sugarcane bagasse tray] × two secondary containers [polyester/polyethylene foam/aluminum foil bag; polypropylene thermobox] × three variants of the number of layers [one layer; three layers; six layers]).

	Variability (%)		
	Total	Explained	*p*
Dish	2	6	0.07
External temperature	18	47	<0.001
Primary wrapping	5	12	<0.001
Secondary container	3	8	<0.01
Number of layers	8	21	<0.001
Dish × Secondary container ^1)^	2	6	0.03
Residual	62		

^1)^ the only significant interaction.

## Data Availability

The data presented in this study are available on request from the corresponding author.

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
