# Peer review of "Key Factors Influencing Bacillus cereus Contamination in Hot Ready-to-Eat Meal Delivery"

_foods, 2025, doi:10.3390/foods14152605_

Round 1

Reviewer 1 Report

Comments and Suggestions for Authors

The manuscript addresses a timely and relevant topic concerning the safety of hot ready-to-eat (RTE) meals during delivery, focusing on temperature retention and Bacillus cereus growth. The experimental design is logical, and the statistical analysis is robust. However, several areas require clarification, methodological justification, and deeper discussion to strengthen the manuscript's impact and validity.

Major Comments:

  1. Methodological Clarifications:

    • Natural Contamination vs. Inoculation: The study relies on natural contamination for B. cereus quantification, which introduces variability in initial spore levels. Please provide data on baseline B. cereus counts in meals before incubation. If unavailable, acknowledge this as a limitation.
    • Temperature Selection for Incubation: The use of 50°C and 60°C for B. cereus growth experiments is unconventional, as B. cereus typically does not grow above 48°C. Justify these temperatures or clarify if they were intended to assess spore survival/thermal resistance rather than growth.
    • Heat-Sealing Temperature: Polypropylene (PP) trays have a melting point near 160°C. Sealing at 180°C may compromise tray integrity. Verify whether container deformation occurred and how this was addressed.
  2. Critical Temperature Threshold:
    The choice of 40°C as the critical temperature for B. cereus growth is not fully justified. While it aligns with safety margins, the manuscript should reference guidelines (e.g., FDA, EFSA) supporting this threshold. Additionally, discuss implications of temperatures between 40°C and 48°C (the upper growth limit for B. cereus).

  3. Toxin Analysis:
    B. cereus pathogenicity is toxin-mediated. The absence of toxin quantification (e.g., cereulide or enterotoxins) limits the safety conclusions. At minimum, acknowledge this limitation and recommend toxin analysis in future work.

  4. Real-World Delivery Conditions:
    The simulated delivery uses static incubation temperatures, whereas real-world conditions involve fluctuations. Discuss how variable external temperatures (e.g., during transit in vehicles) might affect the results and microbial safety.

  5. Packaging Material Trade-Offs:
    While PP trays outperformed sugarcane bagasse (SB) trays in thermal retention, environmental and chemical safety trade-offs (e.g., biodegradability, potential leaching of PP additives) are mentioned but not analyzed. Expand the discussion to balance thermal performance with sustainability/chemical safety concerns.

Minor Comments:

  1. Statistical Reporting:

    • Specify post-hoc test results (e.g., Tukey’s/Nemenyi) in the text or tables, not just p-values.
    • Clarify how the "time interval of temperature decrease to 40°C" was calculated (e.g., regression modeling, direct measurement).
  2. B. cereus Identification:
    Confirm that MALDI-TOF MS differentiated B. cereus sensu stricto from other members of the B. cereus group (e.g., B. thuringiensisB. cytotoxicus), as misidentification could affect conclusions.

  3. External Temperature Ranges:
    The external temperature range (-2°C to 33°C) is broad. Stratify results by temperature categories (e.g., freezing vs. hot conditions) to highlight practical implications for different climates/seasons.

  4. Grammatical/Formatting Issues:

    • Standardize formatting of °C and statistical notation (e.g., "p < 0.05").

Author Response

The authors appreciate very much the effort of the reviewers to improve the manuscript.

We addressed all reviewersÕš comments:

Reviewer #1

Major Comments:

Methodological Clarifications:

Comment: Natural Contamination vs. Inoculation: The study relies on natural contamination for B. cereus quantification, which introduces variability in initial spore levels. Please provide data on baseline B. cereus counts in meals before incubation. If unavailable, acknowledge this as a limitation.

Answer: B. cereus counts in all three tested meals were below the limit of detection (<1.7 log CFU·g-1) before incubation as stated in the Results of the revised manuscript (line 320).

 Comment: Temperature Selection for Incubation: The use of 50°C and 60°C for B. cereus growth experiments is unconventional, as B. cereus typically does not grow above 48°C. Justify these temperatures or clarify if they were intended to assess spore survival/thermal resistance rather than growth.

Answer: We discuss this item in the revised manuscript (Discussion, lines 418-431): Hot RTE meals are distributed in such a way as to reach the consumer as quickly as possible at a temperature of at least 60 °C (requirement of the Czech law), which also corresponds to the Food and Drug Administration requirements for maintaining hot foods at temperatures of 140 °F or higher. While a minimum temperature of 60 °C can be achieved when serving hot dishes in catering service establishments, this is practically impossible when distributing hot RTE meals over longer distances, which is associated with longer time intervals and, consequently, with a risk of the growth of undesirable bacteria, including B. cereus. Therefore, irrespective of the fact that the upper limit of the growth temperature for B. cereus was reported to be 48 °C, the temperatures of 40, 50 and 60 °C were chosen in the present study as model temperatures from the following reasons: 60 °C as the limit temperature given by regulations, and temperatures of 40 °C or 50 °C as the values at which the growth of B. cereus sensu lato is still possible.

Comment: Heat-Sealing Temperature: Polypropylene (PP) trays have a melting point near 160°C. Sealing at 180°C may compromise tray integrity. Verify whether container deformation occurred and how this was addressed.

Answer: The sealing temperature was set according to the film supplier's recommendations. The sealing time was only 1 second and there was no deformation either of the tray or the top film: this comment was added to the Materials and Methods section of the revised manuscript (lines 125-127).

Critical Temperature Threshold:
Comment: The choice of 40°C as the critical temperature for B. cereus growth is not fully justified. While it aligns with safety margins, the manuscript should reference guidelines (e.g., FDA, EFSA) supporting this threshold. Additionally, discuss implications of temperatures between 40°C and 48°C (the upper growth limit for B. cereus).

Answer: It relates to the above-mentioned comment concerning “Temperature Selection for Incubation”, where it has been already discussed (extended discussion in the revised manuscript, lines 418-431). The choice of the temperature of 40 °C was made in order to address a possible situation during distribution of hot dishes by courier services. We do not expect the temperature of hot dishes to drop below 40 °C during distribution to customers. The upper temperature limit, on the other hand, is set by Czech food law, and the same temperature (60 °C = 140 °F) is recommended by the FDA.

Toxin Analysis:
Comment: B. cereus pathogenicity is toxin-mediated. The absence of toxin quantification (e.g., cereulide or enterotoxins) limits the safety conclusions. At minimum, acknowledge this limitation and recommend toxin analysis in future work.

Answer: The present experiment did not focus on the detection of B. cereus toxins. We acknowledge the limitation of the study in the section “Conclusions”: lines 550-554 of the revised manuscript. In further experiments, the authors will focus on the detection of virulence genes in isolated B. cereus strains.

Real-World Delivery Conditions:
Comment: The simulated delivery uses static incubation temperatures, whereas real-world conditions involve fluctuations. Discuss how variable external temperatures (e.g., during transit in vehicles) might affect the results and microbial safety.

Answer: We discuss it in the revised manuscript (lines 463-478) as follows: “It is undeniable that variable external temperatures during delivery can affect microbial safety of the hot RTE meals. However, most hot meals are transported over relatively short distances in thermoboxes using, among other things, bicycles, e-scooters and similar contemporary vehicles. In this case, the temperature conditions are almost identical to the simulated delivery used in the present study, which was demonstrated as safe. If e.g. passenger cars are used for transportation in the colder months, the temperature conditions inside the thermoboxes are again similar to simulations in the laboratory surroundings applied in the present experiment for B. cereus culturing. Nevertheless, a decrease of the internal temperature of the hot meals can still occur, which is also apparent from the results of the present experiment. When taking into account the key drop of temperature below 50 °C and the time interval of the meal delivery under the above-mentioned conditions, our study demonstrated safety of the transport itself: the time interval is not sufficient in the most cases to exceed the lag phase of the B. cereus growth. More important from the viewpoint of the food safety is the behaviour of a consumer after the hot RTE meal has been delivered: how quickly is the meal consumed or under which conditions it is stored until consumption.”

Packaging Material Trade-Offs:
Comment: While PP trays outperformed sugarcane bagasse (SB) trays in thermal retention, environmental and chemical safety trade-offs (e.g., biodegradability, potential leaching of PP additives) are mentioned but not analyzed. Expand the discussion to balance thermal performance with sustainability/chemical safety concerns.

Answer: We expanded discussion as recommended in the revised manuscript (lines 364-370 + 373-376; including several additional references, lines 631-645): In terms of sustainability, sugarcane bagasse trays are clearly superior. Bagasse (formerly burned as a waste) is nowadays used to make eco-friendly packaging and disposable tableware, which are the typical examples of upcycling, i.e., using waste to make a new product. Another advantage of bagasse-based products is their compostability. According to the EN 13432:2000 standard, bagasse decomposes in 45-60 days in an industrial composting plant and in a matter of months in a home compost. On the other hand, PP packaging decomposes hundreds of years; Regarding the food safety concerns, sugarcane bagasse trays do not contain toxic substances and are safe from the viewpoint of contact with food, whereas residual toxic contaminants, such as aldehydes, ketones, aromatic hydrocarbons, plasticizers, etc., were reported to be released from PP packaging into food (lines 365-378 of the revised manuscript).

Minor Comments:

Statistical Reporting:

Comment: Specify post-hoc test results (e.g., Tukey’s/Nemenyi) in the text or tables, not just p-values.

Answer: We specified post-hoc tests as recommended in the text of the revised manuscript. As far as relevant Figures are concerned, specifications of the tests were already present in the original version of the manuscript; Figures 3 and 4, including an accompanying text, are based on regression analysis (there are no post-hoc tests); data presented in Table 1 are calculated based on factorial ANOVA (no post-hoc tests).

Comment: Clarify how the "time interval of temperature decrease to 40°C" was calculated (e.g., regression modeling, direct measurement).

Answer: We added the formulation “The time interval of temperature decrease to the critical value of 40 °C was directly measured” to the section 2.2. of Materials and Methods (line 153).

B. cereus Identification:
Comment: Confirm that MALDI-TOF MS differentiated B. cereussensu stricto from other members of the B. cereusgroup (e.g., B. thuringiensisB. cytotoxicus), as misidentification could affect conclusions.

Answer: The species identification of B. cereus s. s. was performed based on the detection of the gyrB gene encoding the B subunit of DNA-gyrase. Species identification of suspected B. cereus s. l. isolates that were not identified as B. cereus s. s. by PCR was performed using the MALDI-TOF mass spectrometry method. Only isolates with a log (score) above 2.0 were taken into account. To confirm the objectivity of the identification, the following collection strains were used: B. cereus DSM 4312, B. cereus CCM 2010, B. cereus CCM 869, B. thuringiensis CCM 19, B. mycoides CCM 115, B. weihenstephanensis CCM 4872, B. cytotoxicus DSM 22905. We extended in this sense part 2.3. of Materials and Methods in the revised manuscript, including description of the PCR method regarding confirmation of the B. cereus s. s. isolates (lines 186-200).

External Temperature Ranges:
Comment: The external temperature range (-2°C to 33°C) is broad. Stratify results by temperature categories (e.g., freezing vs. hot conditions) to highlight practical implications for different climates/seasons.

Answer: We extended discussion regarding this topic in the revised text (lines 458-462: The experiment was carried out during all seasons of a year within a broad external temperature range from –2 °C to +33 °C. So, it can be concluded in this context that the results of the present study are applicable for both freezing and hot conditions within different seasons and different climate zones.

Grammatical/Formatting Issues:

Comment: Standardize formatting of °C and statistical notation (e.g., "p < 0.05").

Answer: All data regarding °C and statistical notations were standardized in the whole text.

Reviewer 2 Report

Comments and Suggestions for Authors

The paper from Komprda et al. investigates the interesting and very contemporary issue of food safety and meal delivery.

The paper is interesting and well written.

Just some aspects that should be clarified in order to ease readers' understanding:

  • line 109: please better explain why the B cereus growth was assessed in a second aliqout
  • M&M section: is unclear how many samples were prepared
  • Lines: 333-336: please clarify this aspect.
  • Line 391 add a "." at the end of the sentence
  • Line 398: why the maximum time chosen is 4 hours? it seems unlike that a meal delivery would take this long 

Author Response

The authors appreciate very much the effort of the reviewers to improve the manuscript.

We addressed all reviewersÕš comments:

Reviewer #2

Comment: line 109: please better explain why the B cereus growth was assessed in a second aliquot

Answer: We added the explanation into the text concerning the technical difficulties to quantify B. cereus counts in the delivered meals after the each run of the simulated delivery (line 114/115 of the revised text).

Comment: M&M section: is unclear how many samples were prepared

Answer: We specified number of samples in the “Preparation of the dishes” section: n = 119, 67 and 63 as far as mashed potatoes, cooked rice and mushroom sauce, respectively, are concerned; these numbers were used in the part of the experiment concerning simulated meal delivery; for evaluation of B. cereus counts, n = 6 for each of the meal: lines 107-112 of the revised manuscript.

Comment: Lines: 333-336: please clarify this aspect.

Answer: It is possible that the results could be partially affected by the fact that the number of the respective meals put into the sugarcane bagasse trays was nearly ten times lower in comparison with the PP trays: we mention this in the revised manuscript, lines 359/360.

Comment: Line 391 add a "." at the end of the sentence

Answer: We added a full stop at the end of this sentence.

Comment: Line 398: why the maximum time chosen is 4 hours? it seems unlike that a meal delivery would take this long

Answer:  We agree that a time interval of the RTE meal delivery is usually much shorter, but we used the time of 4 h as a possible “worst-case scenario” for the sake of safety; we added this comment in the text of the revised manuscript, lines 451/452.